# Supplements for Smoking-Related Lung Diseases

Naser A. Alsharairi 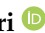

Heart, Mind & Body Research Group, Menzies Health Institute Queensland, Griffith University,
Gold Coast 4222, Australia; naser.alsharairi@gmail.com

**Definition:** Supplements for smoking-related lung diseases are considered as nonfood products and thought to improve health. Multivitamins and antioxidants are the most commonly dietary supplements used by cancer and asthma patients. There are currently no clear regulatory guidelines that include dietary supplements and their effect on lung cancer and asthma patients, particularly in smokers. Several countries have taken steps to overcome challenges in regulating dietary supplements in the marketplace. These challenges include inadequate assurance of safety/efficacy, inaccuracy of product labeling, misleading health claims, and lack of analytical techniques for dietary supplements. There is a need to establish standards and regulation of dietary supplement use in patients with lung cancer and asthma. The aim of this entry is to expand knowledge on dietary supplements use and smoking-related lung diseases (lung cancer and asthma).

**Keywords:** asthma; lung cancer; supplements; smokers; nonsmokers





## 1. Introduction

Smoking is known as one of the main causes of lung cancer and the most common cause of cancer mortality in men and women worldwide [1]. It has been estimated that around 7 million global deaths per year were caused by smoking [2–4]. Cigarette smoke is comprised of thousands of chemical compounds, most of which are toxins [5]. Lung cancer is classified into small cell lung cancer (SCLC) and non-small-cell lung cancers (NSCLCs), with the latter accounting for 85% of lung cancer cases, which is divided into three common subtypes-associated smoking, including large-cell carcinoma, squamous-cell carcinoma, and adenocarcinoma [6]. Reviews of published systematic reviews and meta-analyses have confirmed that the risk of lung cancer is increased in current and former smokers [7–11]. In fact, tobacco smoke is the largest contributor to adenocarcinoma and small-cell and squamous cell carcinoma, with over 76% of lung cancer deaths in men and 37–42% of lung cancer deaths in women aged ≥50 years attributable to tobacco use [12]. The link between smoking and lung cancer risk varies significantly by sex. A number of systematic reviews and meta-analyses examined the sex differences in smoking-related risk of lung cancer. One previous systematic review and meta-analysis showed that currently smoking men had higher susceptibility to lung cancer than women [13]. In a recent meta-analysis, currently/formerly smoking men and women had an increased risk of lung cancer, with no significant sex differences observed [14]. Another recent meta-analysis showed that passive smoking/secondhand smoking (SHS) increased the risk of lung cancer in nonsmoking women [15]. A few studies that have examined the sex differences in histological types of lung cancer have shown that lung adenocarcinoma incidence was higher in women than in men [16,17]. Although a declining prevalence of smoking among women was noted, the risk of mortality from smoking continues to rise [18].

Epidermal growth factor receptor (EGFR) mutations have emerged as a key player in lung tumor development. EGFR stimulates two main downstream signaling pathways; the phosphoinositide 3-kinase (PI3K) and the Ras–Raf–mitogen-activated protein kinase kinase (MEK)–mitogen-activated protein kinase (MAPK) [19]. Kirsten rat sarcoma viral oncogene homolog (KRAS) is identified as the most predominantly mutated oncogene,

with mutations occurring at codons 12 and 13 in NSCLCs [19], particularly among smokers rather than nonsmokers in patients with lung adenocarcinoma [20,21], and affects cellular processes, including tumor angiogenesis, gene transcription, and cell proliferation [22–24]. By contrast, the tyrosine kinase (TK) domain mutations of the EGFR gene are more frequently found in patients with lung adenocarcinoma in nonsmokers than in smokers [25]. Deregulation of EGFR signaling in all histological types may contribute to lung pathogenesis. In fact, aberrant EGFR signaling pathways in most NSCLCs are activated by stimulating EGFR gene mutation, enhancing receptor–ligand binding and increasing the EGFR gene copy number via polysomy/amplification [23,24].

Asthma is one of the major prevalent chronic inflammatory diseases among young adults, the prevalence of which has been shown to be higher in developed countries than developing countries [26], and higher in females than males [27]. Asthma mortality rates were lower than other chronic diseases [28]. However, asthma mortality rates increased with increasing age, and the highest mortality rate was seen in adults aged 65 and over [29]. Asthma mortality data reported that the Netherlands and South Africa had the highest asthma mortality rate globally [26]. Adult-onset asthma may increase the risk of comorbid conditions such as dyspepsia, fluid and electrolyte disorders, chronic obstructive pulmonary disease (COPD), hypertension, congestive heart failure, and diabetes [30].

Asthma is characterized by airway hyperresponsiveness, which results in episodes of shortness of breath (dyspnea), coughing, chest tightness, and wheezing [31]. The type 2 immune responses (Th2) are associated with asthma and mediated by IgE-producing B cells, basophils, type 2 innate lymphoid cells (ILC2s), cytokines, mast cells, and eosinophils [32,33]. A number of environmental factors, including smoking, are associated with the development of asthma in children and adults [34]. Tobacco smoking is reported to cause nearly 10% of asthma mortality worldwide [12]. Tobacco smoke provokes asthma exacerbations and causes other allergy symptoms to worsen in adults [34]. There is also unequivocal evidence that SHS exposure is the main contributor to asthma and lung cancer risk in nonsmokers, disproportionately affecting women [7]. Smoking is able to trigger Th2 inflammation and increase the production of pro-inflammatory cytokines, including interleukin (IL)-4, IL-5, IL-6 IL-13, IL-17A, interferon-$\gamma$ (IFN$\gamma$), and tumor necrosis factor $\alpha$ (TNF-$\alpha$) [34]. A few studies showed higher levels of total immunoglobulin E (IgE) [35,36], sensitive C-reactive protein (hs-CRP), and malondialdehyde (a marker of oxidative stress) [36], in current and passive smokers than never-smokers. Other studies showed increased white blood cell (WBC) counts, and blood monocytes, lymphocytes, neutrophils, and leukocytes in current smokers in comparison to nonsmokers [37,38]. Cigarette smoking stimulates nasal epithelial cells, resulting in increased lipopolysaccharide (LPS) binding, neutrophil chemotaxis, Toll-like receptor (TLR) 4 expression and reactive oxygen species (ROS) production [39]. Cigarette smoking has shown to decrease CD83+ (a surface expression for mature dendritic cells) counts in smoker patients with asthma [40].

Dietary supplement use among adults is rising, but there is a substantial heterogeneity between supplement users depending on the type, consumption frequency, duration, and reason for supplements used [41–43]. Dietary supplement use differs by lifestyle and sociodemographic factors and geographical location among adults [44–47]. Supplement use also varies according to smoking status. Previous studies have shown that ex-and nonsmokers are more likely to be users of supplements than current smokers [45,46].

There is controversy over the role of dietary supplements in reducing or treating lung cancer in smokers and nonsmokers. There is also much uncertainty about its effectiveness and the consequences in asthmatic smokers and nonsmokers, and our understanding of whether dietary supplements can reduce lung cancer risk in asthmatic smokers and nonsmokers remains unclear in the absence of clinical trials [48]. In order to evaluate the safety and effectiveness of dietary supplement use by asthmatic smokers and nonsmokers before, during, and after lung cancer treatment, realistic and reliable studies worldwide are needed.

## 2. Data, Applications, and Influences

Data from clinical trials suggest that specific drugs including RAF/MEK inhibitors failed to demonstrate benefits for patients with oncogene-driven KRAS-mutant NSCLC. For example, treatment of KRAS-mutant lung cancer with selumetinib plus docetaxel [49], trametinib/docetaxel [21,50], or sorafenib [21] failed to show survival benefits or improve response rates. By contrast, RAF inhibition (RO5126766) showed significantly improved response rates in patients with *KRAS*-mutant tumors [51]. Among other inhibitors, the EGFR-TK inhibitors (TKIs) gefitinib, afatinib, erlotinib, osimertinib, and dacomitinib showed clinical benefits in terms of improved progression-free survival and response rates in patients with NSCLCs [52–56].

Dietary supplements are regarded as nonfood but not drug products that are taken to improve health or prevent diseases [57], even though most of the products are unsafe and usually marketed as natural [58]. Dietary supplement use has increased globally [59] and have become of particular interest to consumers and pharmaceutical companies [60,61] in addition to being a significant part of complementary medicines to maintain or improve health [62,63]. Dietary supplements may include multivitamins/minerals, amino acids, herbs, or other botanicals and are available as liquid, tablets, capsules, gelcaps, powder, or softgel [57].The most common dietary supplements used by cancer and asthma patients are multivitamins and antioxidants [64–66]. Globally, the sale of dietary supplements has increased during the years 2011–2016, with the most marked increase in the Asia Pacific region and North America [67].

Different terminologies of the dietary supplements exist, and this results in a supplement being placed in a different category, causing variations in the type of safety assessment and the regulatory frameworks across countries. In Australia and New Zealand, the Food Standards Australia New Zealand (FSANZ) Act treated extracts and foods from animals and plants and modified food substances as "supplements", whereas the Therapeutic Goods Administration (TGA) classified homeopathy, microorganisms (whole extracted), vitamins, minerals, nutritional supplements, and herbs as "complementary medicines". Under the Japanese and Chinese Food Safety Law, the healthy foods including foods for specified health uses/function claims are classified as "supplements", whereas traditional Chinese medicine (e.g., Kampo medicine) is classified as "complementary medicines". In Canada, natural health products including herbal medicine, homeopathy, and traditional medicine are classified as "complementary medicines" [68].

Dietary supplements are the most common forms of alternative therapies to promote health and prevent diseases if used appropriately for those who need them in specific circumstances [69,70]. A healthy balanced diet can contribute to improve micronutrient intake. However, dietary supplements may not offer the desired health benefits, excluding individuals who need to meet nutrient requirements [69]. Dietary supplements could help meet the nutritional needs for individuals at risk of micronutrient deficiencies, but their contribution to the intake of nutrients should be detected by nutritional biomarkers [70].

There is a strong belief that taking dietary supplements prevents occurrence of cancers [71]. To the contrary, these supplements are not safe for cancer patients and lead to increased risk of mortality in Westernized adult populations [72]. Several human experimental studies found negative or no effect from vitamin C, vitamin E ($\alpha$-tocopherol), vitamin A, selenium, folic acid, $\beta$-carotene, or vitamin D/calcium supplementation on risk of cancers such as breast, skin, colorectal, and prostate [71]. The evidence for the role of dietary supplements in the prevention of lung cancer relies on randomized controlled trials (RCT), regarded as the "gold standard". Indeed, RCTs have been contradictory regarding the effects of dietary supplements on lung cancer risk and mortality in smokers and nonsmokers [48]. However, there is now scientific evidence that urges caution in recommending long-term, high-dose supplements that contain $\beta$-carotene, retinyl palmitate, B vitamins and vitamin E for lung cancer patients, particularly current and former smokers [48,73], suggesting that these supplements are likely to harm rather than provide benefits to such patients.

Given that long-term supplement use can have adverse effects, why do companies still claim anticancer benefits in the marketing of these supplements? Pharmaceutical companies have not adequately complied with dietary supplement manufacturing standards. These supplements contain harmful ingredients which continue to be sold to cancer patients for commercial reasons [74]. Sales of dietary supplements could represent harm, rather than health benefits, if national standards and regulations do not rigorously apply. Regulating and monitoring the marketplace is important in countries where dietary supplement use is high, and where certain micronutrient intake is low. Without regulating supplements and monitoring their efficacy or safety, how can lung cancer patients make a decision to use them? Indeed, there are no clear recommendations to use dietary supplements during lung cancer treatment. Few countries have existing regulatory frameworks for dietary supplements. For instance, the US Food and Drug Administration (FDA) established a regulatory framework governing the marketing and manufacturing of dietary supplements, including establishing labeling, packaging, and storage rules for specific products like botanicals, herbals, and vitamin/mineral supplements, defining products of significant value based on the recommended dietary allowance (RDA), implementing regulatory guidelines to improve the marketplace and evaluating the efficacy of dietary supplement ingredients. Under the US Act 1994, products are not required to be gained premarket approval process and assessed for efficacy. Safety concerns have been raised due to adulterants and contaminants in herbal and botanical products used in weight loss, sexual enhancement, and body building. The regulatory barriers for dietary supplements in the US marketplace affect regulator's ability to carry out their roles. These include a significant number of products and companies and a lack of standard methods applied to the analysis of new products [75]. In Europe, the European Food Safety Authority (EFSA) proposed a number of legal and regulatory guidelines for evaluating the toxicity of the source of vitamins/minerals, bioactive compounds (including botanicals), and other substances intended for use as ingredients in fortified foods, novel foods, or food supplements. The toxicity evaluation has been classified into four major areas: reproductive and developmental toxicity, genotoxicity, toxicokinetics, and chronic toxicity/carcinogenicity [76]. In the European Union, safety assessments are needed for new supplement ingredients, which should be approved before marketing. Safety laws aim to use a combination of safety strategies to protect the health of consumers, including analytical approaches and expertise [77]. The EFSA panel has raised concerns about the safety and toxicity of chromium picolinate used in food supplements and reported that a high daily dose of chromium should be re-evaluated in the future [78]. The panel has reported that the presence of pyrrolizidine alkaloids (PAs) in herbal dietary supplements and herbal infusions act as genotoxic carcinogens in humans and could cause severe acute toxicity. Therefore, the panel recommended performing risk assessment of PAs in these supplements [79]. The panel has also performed a tiered approach to toxicological assessment of sources of nutrients proposed for inclusion in dietary supplements. This approach integrates the areas of developmental and reproductive toxicity, carcinogenicity, genotoxicity, and kinetics [80]. Countries like Australia, Canada, China, and Japan have proposed that incorporating new technologies such as DNA barcoding, an omics-based approach, blockchain, in silico tools, and advanced analytics techniques may improve guidelines on safety evaluation and quality control of dietary supplements [68].DNA barcoding is a powerful technique which detects deceptive dietary supplements [68]. In North America, herbal dietary supplement fraud may occur due to product substitution with cheap/contaminated species and fraudulent labeling [81]. An omics-based analytical approach is a novel tool for assessing the safety of dietary supplements such as botanical and herbal products [68]. Transcriptomics is among the most used omics technologies in dietary supplements research. In China, transcriptomics using DNA microarray with high reproducibility and reliability of the data has been applied to assess the safety of herbal medicine [82]. Blockchain technology proposes a data digital system using reliable cryptographic algorithms to enable users to monitor the quality assurance of dietary supplement and herbal medicinal products [68]. In China and India, product purity and quality,

including levels of heavy metal contamination, are controlled using blockchain [83]. In silico chemical safety assessment is one type of toxicity assessment that uses computational modeling to improve safety assurance and toxicity prediction of the botanical and dietary supplements. This approach takes advantage of identifying the chemical constituents of botanicals that cause hazards and unresolved safety endpoint gaps [84]. Among advanced analytical techniques, next-generation sequencing has emerged as an effective reliable tool for taxonomic authentication of herbal supplements, which has a significant role to play in detecting fungal DNA and evaluating the purity of the final product [85].

E-cigarettes and e-liquids are now sold as dietary supplements for therapeutic purposes [86]. Several tobacco and cigarette companies market e-cigarettes as dietary supplement products and claim they are safe and offer health benefits. For instance, Vapes claims on its website (https://www.vapes.com) that its products are nicotine-free and contain peach berry e-liquid associated with weight management aids. NutroVape's website (https://nutrovape.com) claims that its products contain green tea extract, garcinia cambogia, L-theanine, *Hoodia gordonii*, and natural passionflower/chamomile associated with improved sleep and appetite reduction. VitaCig (https://vitaciggroup.com) claims that their e-cigarette devices deliver vitamins, such as coenzyme Q10 and vitamins C, A, B12 and E, and essential oil inhalation, such as of menthol, which can help control appetite [87]. However, these products may not have any therapeutic efficacy to be used for treating pathological conditions such as lung cancer and asthma. E-cigarette and products containing nicotine should not be treated as dietary supplements. This is because dietary supplements do not contain any active pharmaceutical ingredients that are approved as prescription medications before being marketed as foods or supplements. A number of adulterated dietary supplements contain unapproved pharmaceutical ingredients under the US FDA Act such as fluoxetine [88], which is used as an antidepressant in smokers [89] but suspected to be carcinogenic [90].

RCTs have revealed that supplementation with dietary supplements, including vitamin B6 and choline, were found to improve pulmonary function and decrease asthma symptoms, allergies, atopy, and serum IgE in patients with asthma. However, no significant association was found between vitamin B12 supplementation and a reduced risk of asthma or wheeze [91,92]. Pharmaceutical companies are involved in manufacturing supplements purported for use in asthma such as quercetin and pycnogenol [93]. Quercetin is one of the most natural polyphenolic antioxidants found in many food products, which exerts anti-allergic effects by suppressing airway hyperresponsiveness and eosinophil chemoattractants induced by periostin, leading to reduced risk of asthma and allergic diseases [94]. Pycnogenol (French *Pinus pinaster*) is a natural plant extract contains polyphenolic compounds capable of reducing asthma by inhibiting pro-inflammatory cytokine production [95]. A need remains for more efforts to provide regulatory guidelines about using dietary supplements to reduce asthma risk. Nicotine-containing e-cigarette aerosols have been reported to impair lung function [96]. Vaping e-liquids cause centrilobular nodules and ground-glass opacities in lung imaging, and have the potential to cause death [97]. There is concern as prior studies suggest that e-cigarettes are most often used by adolescents with asthma [98–100]. The basic components of e-cigarettes aerosols/e-liquids include flavoring agents, metals, drugs, aldehydes, tobacco alkaloids, phenolic compounds, nicotine, glycerol, and propylene glycol (PG), which contain potentially harmful toxicants (e.g., nitrosamines, acetaldehyde, acrolein, formaldehyde) contribute to the pathogenesis of asthma [96]. Glycerol is commonly used as food additives and reported to be related to the presence of toxicological compounds such as acrolein and 3-monochloropropane-1,2-diol (3-MCPD) [101]. Flavorings used in e-liquids have harmful effects, causing oxidative stress and inflammation in lung epithelial cells [102]. Natural plant extracts and cinnamon-flavored e-cigarette are reported to induce pro-inflammatory mediator interleukin-8 (IL-8) and impair respiratory barrier function [103]. Aldehydes and acrolein, used as food flavoring agents, are potent toxins causing adverse health effects including chronic noncommunicable human diseases (NCDs) (e.g., neurological or

cardiovascular diseases, cancer) [104]. Cinnamaldehyde in flavored e-cigarette liquids is found to induce suppression of ciliary beat frequency (CBF), a major factor in the defense of the respiratory tracts, which may lead to increased risk of lung diseases including asthma [105]. This suggests that flavored e-cigarettes may increase the risk of asthma. Few RCTs to date have sought to examine the effects of dietary supplements on asthma risk in smokers. Although vitamin D supplementation alone has proven beneficial in reducing asthma risk in current/former smokers, intake of vitamin D supplements together with calcium/other supplements may not be advocated [48]. Further RCTs to examine the efficacy of dietary supplements in improving asthma symptoms are needed [106]. More RCTs are also needed to clarify the effects of dietary supplements on asthma risk in both smokers and nonsmokers.

## 3. Conclusions

Dietary supplement use is popular among adult patients with cancers in many countries, due to the belief that these products are safe and will improve health or prevent cancers. Active and passive smoking are the main risk factors for lung cancer and asthma. People with asthma are generally at higher risk of lung cancer than the general population, whether they smoke or are exposed to tobacco smoke. Dietary supplement use in lung cancer prevention elicits considerable controversy. Long-term use of specific dietary supplements appears to increase lung cancer risk among current smokers. Although vitamin D supplementation has limited benefits in reducing asthma, dietary supplements have, in general, failed to prove that they are safe or effective on lung cancer prevention among current/former smokers. Dietary supplements might not actually reduce the risk of other cancers because many studies have made very grand claims.

The naturopathy sector makes millions of dollars by making claims about cancer-fighting supplements, but these should be backed up with empirical research, and if false, those companies should not be profiting from misleading people. The quality, efficacy, and safety of dietary supplements are among the greatest potential regulatory challenges that exist in many countries, but what are the recommendations for further work in this field? Appropriate regulatory oversight for evaluating and monitoring the quality and safety of dietary supplements targeting lung cancer and asthma patients is needed before they are distributed to the market. Enforcing regulatory rules may improve techniques to assess the bioactivity, quality, safety, and purity of dietary supplements. Improving the accuracy of nutrient and chemical measurements in dietary supplements would be needed to ensure that these products are safe before being used by lung cancer and asthma patients. Strict rules would also be needed to identify a safe dosage range of dietary supplements used by lung cancer and asthma patients, particularly in smokers. A need remains for implemented new rules to eliminate or restrict use of flavored e-cigarettes. Manufacturing companies need to comply with the regulations, guidelines, and policies governing the marketing of dietary supplements, and should be banned from manufacturing and marketing new products that pose substantial health risks, particularly via products targeting cancer and asthma patients. Since most dietary supplements are unsafe products, it is recommended that lung cancer and asthma patients consult health professionals, including physicians, prior taking these products. The mechanisms of action underlying the effects of dietary supplements on asthma and lung cancer risk in smokers and nonsmokers remain undetermined. Given that active and passive smoking are the main risk factors for asthma and lung cancer, future trials and prospective studies of dietary supplements to evaluate their therapeutic mechanisms of action for lung cancer prevention in asthmatic smokers and nonsmokers are needed.

**Funding:** This entry received no financial support.

**Conflicts of Interest:** The author declares no conflict of interest.

**Entry Link on the Encyclopedia Platform:** https://encyclopedia.pub/6926.

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
