# Peer review of "Supplements for Smoking-Related Lung Diseases"

_encyclopedia, doi:10.3390/encyclopedia1010010_

Round 1

Reviewer 1 Report

I am satisfied with the revision. 

Author Response

Thank you

Reviewer 2 Report

Dear Author,

I think the changes you have made have vastly improved the manuscript. Thank you for your efforts. I have outlined a few minor points below for your attention. I do not mean to be critical, I simply hope to help you get the best out of your efforts.

Line 13 – remove mention of the researchers and journal considering you are talking about yourself. There is no need for the third person approach. Like you did in line 98.

I don’t mean that you can’t cite it or use its material. I just mean that this has to read like an encyclopedia entry (certainly a lot better with your changes). An encyclopedia entry  is informative, to the point and doesn’t ever mention researchers or specific studies in the text… unless a particular researcher or famous study is the target of the entry. For example Albert Einstein – General Relativity. You are nearly there I believe.

Line 25 – it would be nice to have a follow-up statistic on maybe smoking and asthma or another smoking related disorder like the skin conditions etc. that can occur just to reinforce your point that it is a major health issue

Line 88-93 and 147-153 great points made. Always important to have perspective.

Line 191-197 – can you please cite the website url of these companies as a reference, would be great for people to see examples.

Line 205 – somewhere around here or later on int the text you should mention that vaping causes ground glass opacities in lung imaging, and even has  caused some deaths.

Best of luck!

Author Response

Dear Reviewer

Thank you for your comments

Dear Author,

I think the changes you have made have vastly improved the manuscript. Thank you for your efforts. I have outlined a few minor points below for your attention. I do not mean to be critical, I simply hope to help you get the best out of your efforts.

Line 13 – remove mention of the researchers and journal considering you are talking about yourself. There is no need for the third person approach. Like you did in line 98. I don’t mean that you can’t cite it or use its material. I just mean that this has to read like an encyclopedia entry (certainly a lot better with your changes). An encyclopedia entry is informative, to the point and doesn’t ever mention researchers or specific studies in the text… unless a particular researcher or famous study is the target of the entry. For example Albert Einstein – General Relativity. You are nearly there I believe.

Response: I've removed mention of the researcher and journal.

Line 25- it would be nice to have a follow-up statistic on maybe smoking and asthma or another smoking related disorder like the skin conditions etc. that can occur just to reinforce your point that it is a major health issue.

Response: I've mentioned this in Line 70.

Line 88-93 and 147-153 great points made. Always important to have perspective.

Response: Thank you for your comment.

Line 191-197 – can you please cite the website url of these companies as a reference, would be great for people to see examples.

Response: I've added the website url of these companies (Line 205-210).

Line 205 – somewhere around here or later on in the text you should mention that vaping causes ground glass opacities in lung imaging, and even has caused some deaths.

Response: I've mentioned this in Line 231-232.

This manuscript is a resubmission of an earlier submission. The following is a list of the peer review reports and author responses from that submission.

Round 1

Reviewer 1 Report

The authors’ topic review filled the gap between dietary supplements and cancer risk in asthmatic smokers/nonsmokers. It was well written. Please see two minor comments below.

Major points

  1. Smoking increases lung cancer but not all types of lung cancer. For example, some lung cancer patients harboring EGFR mutations do not smoke. This limitation of the link between smoking and lung cancer risk should be highlighted in the main text.

Minor points

  1. The author should mention that the title of the Heading 3.1 “Smoking and Risk of Asthma” (PMID: 30925812) is misleading. In the paragraph the effect of vitamin D on lung function is discussed; therefore, the title should be changed accordingly. 
  2. Most of the headings need to be revised as they do not give an accurate representation of the material to be discussed in the following paragraphs. Maybe the titles should include the dietary supplement that is going to be discussed. 

Reviewer 2 Report

In the text “Asthma, Lung Cancer and Supplements”, the author described the risk factors for asthma and lung cancer and the effect of dietary supplements on lung cancer risk/mortality in asthmatic smokers and nonsmokers. This entry is very interesting and may contribute to the knowledge in the field of life science. Generally, the text is well written and the composition is easy to understand.

Reviewer 3 Report

According to the guidelines put forth by Encyclopaedia the aim of an Encyclopaedia entry is “to form a comprehensive record of scientific development as well as provide reference information for researchers and the general public who are interested in accurate and advanced knowledge on specific topics”. The authors have chosen to do a topic review in which “. . . scholars can: (1) introduce new discoveries or concepts, introduce the main concepts and ideas, which may also include the origin, development history, milestones, leading figures or related people, current status, predictions, future directions, and any other relevant information; (2) present the conclusions of one or several research projects on a single, specific topic, or a general review of a research field. Results can relate to a single field, a research group, or a collective of researchers; (3) provide research project proposals and expected results.”.

While the piece is well written, the encyclopaedia entry by Alsharairi does not seem to follow this approach and seems to be more akin to a summary of a review that they themselves conducted. As the piece they are referring to happens to be a literature review, it cannot be considered a scientific development or a project with expected outcomes per se as it is not an original research study. As a result the piece lacks critical substance that might be useful to someone researching this field.  

Strangely, throughout the entry the author keeps referring to themselves in the third person even though it is their own research they are presenting? There is no need for that.

Indeed, the first mention of a specific type of supplement does not come until line 47 (out of a total of 64).  

While the review in the journal Nutrients is one that I have read and find very interesting, I would refer people to the review. I would not refer people to a review of a review. Therefore, I this does not constitute an Encyclopedia entry.